# The Role of ICG in Robot-Assisted Liver Resections

**DOI:** 10.3390/jcm11123527

**Published:** 2022-06-19

**Authors:** Anne-Sophie Mehdorn, Florian Richter, Katharina Hess, Jan Henrik Beckmann, Jan-Hendrik Egberts, Michael Linecker, Thomas Becker, Felix Braun

**Affiliations:** 1Department of General, Abdominal, Thoracic, Transplantation and Pediatric Surgery, University Hospital Schleswig-Holstein, Campus Kiel, Arnold-Heller-Straße 3, 24105 Kiel, Germany; anne-sophie.mehdorn@uksh.de (A.-S.M.); florian.richter@uksh.de (F.R.); jan.beckmann@uksh.de (J.H.B.); michael.linecker@uksh.de (M.L.); thomas.becker@uksh.de (T.B.); 2Department of Pathology, University Hospital Schleswig-Holstein, Campus Kiel, Arnold-Heller-Straße 3, 24105 Kiel, Germany; katharina.hess@uksh.de; 3Department of Surgery, Israelit Hospital, Orchideenstieg 14, 22297 Hamburg, Germany; j.egberts@ik-h.de

**Keywords:** robotic surgery, robot-assisted hepatic surgery, robotic hepatic surgery, liver surgery, robotic liver surgery, hepatic surgery, minimally invasive hepatic surgery, ICG, indocyanine green

## Abstract

Introduction: Robotic-assisted liver surgery (RALS) with its known limitations is gaining more importance. The fluorescent dye, indocyanine green (ICG), is a way to overcome some of these limitations. It accumulates in or around hepatic masses. The integrated near-infrared cameras help to visualize this accumulation. We aimed to compare the influence of ICG staining on the surgical and oncological outcomes in patients undergoing RALS. Material and Methods: Patients who underwent RALS between 2014 and 2021 at the Department of General Surgery at the University Hospital Schleswig-Holstein, Campus Kiel, were included. In 2019, ICG-supported RALS was introduced. Results: Fifty-four patients were included, with twenty-eight patients (50.9%) receiving preoperative ICG. Hepatocellular carcinoma (32.1%) was the main entity resected, followed by the metastasis of colorectal cancers (17%) and focal nodular hyperplasia (15.1%). ICG staining worked for different tumor entities, but diffuse staining was noted in patients with liver cirrhosis. However, ICG-supported RALS lasted shorter (142.7 ± 61.8 min vs. 246.4 ± 98.6 min, *p* < 0.001), tumors resected in the ICG cohort were significantly smaller (27.1 ± 25.0 mm vs. 47.6 ± 35.2 mm, *p* = 0.021) and more R0 resections were achieved by ICG-supported RALS (96.3% vs. 80.8%, *p* = 0.075). Conclusions: ICG-supported RALS achieve surgically and oncologically safe results, while overcoming the limitations of RALS.

## 1. Introduction

Presently, robotic procedures, including hepatic procedures, are being increasingly performed [1,2,3]. The advantages of robotic surgery are well known, and robotic liver surgery has been proven to be safe [4,5,6]. Despite advantages such as steady and clear 3D visualization, superior ergonomics, and improved dexterity, a major limitation of robotic surgery is its lack of haptic feedback, so much so that the surgeon purely relies on the visual aspects [1,7,8]. In open hepatic surgery, an intraoperative ultrasound (IOUS) is mandatory to add precision and allow real-time intraoperative, strategic imaging [9]. Even with robot-assisted liver surgery (RALS), IOUS is an additional tool that may help to overcome the lack of the haptic feedback during such robotic procedures [10].

Another tool to add precision and help perform more targeted hepatic surgery is the drug indocyanine green, ICG, which was Federal Drug Administration (FDA) approved in 1954 and has been used extensively in different medical fields ever since [11,12,13]. In hepatic surgery, ICG is used not only for the intraoperative real-time visualization of bile ducts and blood flow, but also for the differentiation of cancerous and normal tissue, depending on the manner of application (positive vs. negative staining) [13,14]. Furthermore, it is used to test liver function prior to major hepatic resections (Limon test) [11]. In this context, Ishizawa et al. discovered persisting ICG staining in patients who had received ICG days before hepatic surgery [15]. ICG staining, either as an intra-tumoral accumulation in hepatocellular carcinoma (HCC) or a halo-like rim accumulation in colorectal liver metastasis (CRLM), was visualized by near-infrared light at an approximate wavelength of 835 nm, presumably due to impaired biliary excretion mechanisms and congestion in perimetastatic cells [11,13,15,16,17].

The feasibility of intraoperative ICG staining and its usage as an additional intraoperative tool for HCC and liver metastasis (LM) have been reported by different groups [1,8,10,17,18,19]. Nowadays, some laparoscopic and robotic systems possess an integrated near-infrared camera that allow intra-operative, real-time visualization of ICG accumulation, thus providing real-time information on the extent of the tumor or LM, respectively, to the operating surgeon who can adapt the intraoperative strategy according to the visualized staining. [10,15,20].

Despite its long-standing approval and use, ICG presents itself with disadvantages as it needs to be applied intravenously and comes with the risk of complications during the application, i.e., paravasation, allergic reaction to iodine or any other solubles, as well as hyperthyroid crisis in patients suffering from unknown hyperthyroidism [13,21].

With this study, we, therefore, aim to investigate whether ICG provides helpful additional information to the surgeon and improves the surgical and oncological outcome of patients undergoing RALS for HCC and LM of different origins, respectively. We further want to investigate whether the ICG-supported RALS is associated with an increased occurrence of side effects and possibly prolongs operation time without improving the oncological outcome.

For this purpose, a comparison of intra- and postoperative courses of patients who received ICG preoperatively (ICG-supported RALS) and a historical comparison cohort (RALS) from our center without preoperative ICG administration was performed.

## 2. Material and Methods

### 2.1. Study Design and Study Population

Patients who underwent RALS between 2014 and 2021 at the Department of General, Visceral, Thoracic, Transplantation, and Pediatric Surgery, University Hospital Schleswig-Holstein, Campus Kiel, were included. All patients received preoperative radiological imaging, i.e., a magnetic resonance imaging (MRI) or computer tomography (CT) scan. The inclusion criterion was eligibility for RALS. The exclusion criteria were conversion of the procedure or combination with another operation other than simultaneous cholecystectomy.

ICG was introduced for RALS at the department in 2019. The additional exclusion criteria following its introduction were iodine allergy and hyperthyroidism. ICG-supported RALS (ICG cohort: patients receiving preoperative intravenous ICG, between 2019 and 2021) was compared to a historic cohort (HC) (patients who underwent RALS between 2014 and 2019) who did not receive preoperative ICG.

Patient data were retrieved from a prospectively maintained database (2019–2021) and compared to the data of HC. Only de-identified data were used according to the Declaration of Helsinki. All patients had given their informed consent to the participation in clinical trials, and the local ethics committee had given their approved consent (D 610/20). Patient data included demographic, as well as surgery-specific and oncological data.

### 2.2. ICG Application and Evaluation

ICG (25 mg, Verdye^®^, Diagnostic Green GmbH, Aschheim, Germany) was dissolved in sodium hydrochloride or aqua according to the operator’s manual and applied preoperatively intravenously immediately after dilution, following the exclusion of iodine allergy or hyperthyroidism. Intraoperative ICG staining was evaluated postoperatively by the operating surgeon using a specifically created in-house questionnaire.

### 2.3. Surgery

RALS was performed as previously described using the Da Vinci Robotic Si^TM^ and Xi^TM^ Systems (Intuitive Surgical Inc.^®^, Sunnyvale, CA, USA) [8,22].

### 2.4. Histological Analysis

All specimens were surgically collected, fixed in formalin and embedded in paraffin. Then, 5 µm thick sections were routinely processed with hematoxylin and eosin (HE), periodic acid–Schiff reaction (PAS), iron, chromaniline blue, and Gomori stains. The diagnosis of HCC was performed according to the current World Health Organization (WHO) classification [23].

### 2.5. Outcome Measures

The primary endpoint of this study was the feasibility of RALS. The secondary endpoints were surgery-specific and oncological outcomes. In the ICG cohort, ICG application and ICG-supported RALS were the additional secondary endpoints.

### 2.6. Statistical Analysis

The qualitative data are presented as means ± standard deviation (SD). Data were compared either using Students’ *t*-test (normally distributed) or Mann–Whitney U test (non-normally distributed). Quantitative data were presented as percentage and compared using chi-square or Fisher’s exact test. Statistics were performed using GraphPad Prism version 9.3.1 (350) (GraphPad Software, San Diego, CA, USA) and SPSS 28.0.0.0 (190) (IBM, Armonk, NY, USA) for Mac.

## 3. Results

During the inclusion period, sixty-seven patients underwent RALS at the Department of General, Abdominal, Thoracic, Transplantation, and Pediatric Surgery, University Hospital Schleswig-Holstein, Campus Kiel, including one patient who underwent two RALS during the inclusion period. Thirteen patients were excluded from the final analysis due to the previously mentioned exclusion criteria, including nine patients (11.9%) who underwent conversion to an open procedure, due to bulky or centrally situated tumors. Thus, the final analysis included 54 patients. The patients’ average age was 64.0 ± 14.3 years, and 50.0% of the participants were male (Table 1, first column from the left). Tumors were mainly HCC (32.1%) and CRLM (17.0%), followed by focal nodular hyperplasia (FNH) (15.1%). Histopathologically proven liver fibrosis and cirrhosis were noted as 9.26% and 24.1%, respectively. The tumors resected had an average diameter of 36.9 ± 31.8 mm, with a distance of 7.8 ± 12.1 mm to the resection margin (RM). Overall, intra- and post-operative complication rates were low (5.6% and 16.7%, respectively) (Table 2).

Twenty-eight (50.9%) patients had received preoperative intravenous ICG. The cohorts were defined accordingly, i.e., ICG (*n* = 28) vs. HC (*n* = 26), respectively. In one patient, we noticed a paravasation due to a misplaced intravenous line, which did not cause further morbidity. No ICG-associated adverse events were noted pre- and post-operatively.

The cohorts were well-comparable with regard to the demographic and clinical data (Table 1, middle and right columns). A total of 50.0% and 38.5%, respectively, had undergone previous surgery (*p* = 0.394) with sigma- and low anterior resections (21.4% vs. 23.1%, *p* = 0.918), appendectomies (7.1% vs. 30.8%, *p* = 0.114), and cholecystectomies (21.4% vs. 0.0%, *p* = 0.077), respectively, being the leading procedures in both cohorts.

**Table 1 jcm-11-03527-t001:** Demographic baseline comparison of patients who underwent RALS with and without obtaining preoperative intravenous ICG.

	Total (*n* = 54)	ICG (*n* = 28)	HC (*n* = 26)	*p*-Value *
**Clinical Data**				
Age (years) mean ± SD	64.0 ± 14.3	65.4 ± 11.1	62.5 ± 17.1	0.465 ^a^
Sex (males) %	50.0	53.6	46.2	0.586 ^b^
BMI (kg/m^2^) mean ± SD	27.5 ± 5.3	28.7 ± 7.0	26.8 ± 4.3	0.389 ^a^
Liver fibrosis (histopathologically proven)	13.0	14.3	11.5	0.764 ^b^
Liver cirrhosis (histopathologically proven)	24.1	17.9	30.8	0.264 ^b^
Previous abdominal surgery (yes) %	44.4	50.0	38.5	0.394 ^b^
Type of previous surgery (open vs. MI) %	38.5/61.5	21.4/78.6	58.3/41.7	0.054 ^b^
Open	38.5	21.4	58.3	0.054 ^b^
Minimally-invasive	30.8	42.9	16.7	0.149 ^b^
Robot-assisted	30.8	35.7	25.0	0.555 ^b^
Previous procedure				
Sigma/LAR	22.2	21.4	23.1	0.918 ^b^
Appendectomy	18.5	7.1	30.8	0.114 ^b^
Cholecystectomy	11.1	21.4	0.0	0.077 ^b^
Herniotomy	7.4	7.1	7.7	0.957 ^b^
Right nephrectomy	7.4	7.1	7.7	0.957 ^b^
Cystoprostatectomy	7.4	14.3	0.0	0.157 ^b^
Caesarean section	7.4	0.0	7.7	0.290 ^b^
Gynecological surgery	7.4	0.0	15.4	0.127 ^b^
Esophagectomy	3.7	7.1	0.0	0.326 ^b^
Right hemicolectomy	3.7	7.1	0.0	0.326 ^b^
Left hemicolectomy	3.7	0.0	7.7	0.290 ^b^
Adrenalectomy	3.7	7.1	0.0	0.326 ^b^

Data are presented as mean ± standard deviation (SD), range, or relative frequencies. Continuous variables were tested using ^a^ Students’ *t*-test (normally distributed), while categorical variables were compared using ^b^ Chi-square; “* *p*-values” refer to values in the middle and right column. BMI: body mass index; LAR: low anterior rectum resection; ICG: indocyanine green; HC: historic cohort; MI: minimally invasive and SD: standard deviation.

In both cohorts, segment (44.4% vs. 26.9%, respectively, *p* = 0.221) and wedge (40.7% vs. 26.9%, *p* = 0.513) resections were the leading procedures (Table 2, middle and right column). Tumors and metastasis were mainly located in segments II, V, VI, and VIII in the ICG cohort, while segments II and III were the main segments involved in the procedures in the HC. Intraoperative complications were realized for 10.8% in the ICG cohort. They consisted of venous bleeding in two cases and a bile leak in another case. All complications were fully treated robotically without warranting conversion or any further action. No intraoperative complications were reported in HC. However, the reliability of this is highly debatable. Of note, the procedures lasted significantly longer in HC (142.7 ± 61.8 min vs. 246.4 ± 98.6 min, *p* < 0.001). Longer procedure times in HC may be ambulated by the significantly larger tumors resected (27.1 ± 25.0 mm vs. 47.6 ± 35.2 mm, *p* = 0.021), significantly more simultaneous cholecystectomies (25.0% vs. 58.3%, *p* = 0.030), as well as the performance of major liver resections; especially, more right hemi-hepatectomies (0.0% vs. 11.5%, *p* = 0.031) were observed.

**Table 2 jcm-11-03527-t002:** Post-operative data and histological results of the comparison of patients who underwent RALS with and without obtaining preoperative intravenous ICG.

	Total (*n* = 54)	ICG (*n* = 28)	HC (*n* = 26)	*p*-Value *
**Post-operative Data**				
Post-operative complications (yes) %	16.7	14.3	19.2	0.636 ^b^
Clavien–Dindo I	11.3	10.7	11.5	0.923 ^b^
Clavien–Dindo II	5.7	3.6	7.7	0.486 ^b^
Clavien–Dindo III	0.0	0.0	0.0	na
Clavien–Dindo IV	1.9	3.6	0.0	0.331 ^b^
Clavien–Dindo V	0.0	0.0	0.0	na
Size of tumor (mm) mean ± SD	36.9 ± 31.8	27.1 ± 25.0	47.6 ± 35.2	**0.021 ^a^**
Histopathological results				
Hepatocellular carcinoma	32.1	29.6	34.6	0.633 ^b^
Hepatocellular carcinoma in cirrhosis	18.5	14.3	23.1	0.406 ^b^
Colorectal cancer	17.0	22.2	11.5	0.330 ^b^
Focal nodular hyperplasia	15.1	11.1	19.2	0.379 ^b^
Cholanciocellular carcinoma	7.3	0.0	15.3	**0.015 ^b^**
Haemangioma	5.7	0.0	11.5	0.135 ^b^
Breast cancer	3.8	7.4	0.0	0.165 ^b^
Neuroendocrine tumor	3.8	7.4	0.0	0.165 ^b^
Gastrointestinal stroma tumor	1.9	0.0	3.8	0.295 ^b^
Anal cancer	1.9	0.0	3.8	0.295 ^b^
Choroid coat melanoma	1.9	3.7	0.0	0.331 ^b^
Non-small cell lung cancer	1.9	3.7	0.0	0.331 ^b^
Leiomyosarcoma	1.9	3.7	0.0	0.331 ^b^
No malignancy proven	5.7	11.1	0.0	0.086 ^b^
Distance to resection margin (mm) mean ± SD	7.8 ± 12.1	5.8 ± 10.9	10.2 ± 13.4	0.200 ^a^
Resection margin positive-resections (yes) %	11.3	3.7	19.2	0.075 ^b^
Length of hospital stay (days) mean ± SD	6.4 ± 4.0	5.9 ± 5.0	6.9 ± 2.7	0.383 ^a^

Data are presented as mean ± standard deviation (SD), range, or relative frequencies. Continuous variables were tested using ^a^ Students’ *t*-test (normally distributed), while categorical variables were compared using ^b^ Chi-square; values in bold were considered statistically significant (*p* < 0.05). “* *p*-values” refer to values in the middle and right column. ICG: indocyanine green; HC: historic cohort; SD: standard deviation.

Most tumors resected were HCC (29.6% vs. 11.5%, *p* = 0.330) and CRLM (22.2% vs. 11.5%, *p* = 0.379) (Table 3). Postoperative complications were comparable between both cohorts (14.3% vs. 19.2%, *p* = 0.636). Despite the distance to resection margin in HC being wider (5.8 ± 10.9 mm vs. 10.2 ± 13.4 mm, *p* = 0.200), there was a tendency toward more R1-resections in the latter cohort (3.7% vs. 19.2%, *p* = 0.075), with tumors in the HC being significantly larger (27.1 ± 25.0 mm vs. 47.6 ± 35.2 mm, *p* = 0.021 (Table 3).

Successful ICG staining was noticed in 22 patients who had received 0.31 ± 0.06 mg ICG/kg bodyweight 19:86 ± 24:42 h prior to the surgery. In these patients, the intraoperative ICG staining was considered helpful, clear, and unequivocal in 81.8% of the participants. Surgeons who performed ICG-supported RALS rated the intraoperative ICG staining to be very good (1.68 ± 0.64; ranging from 1 = very good to 6 = very poor). In 90.0% of the ICG-assisted RALS, an IOUS was used. A correlation between the two tools, ICG and IOUS, was noticed in 81.0%. The combination of both tools was considered most helpful by the performing surgeons (45.5%). Staining signals were, as previously described, with total tumor-staining in HCCs, partial fluorescence and halo phenomenon in metastasis, and diffuse-staining in CCC (Figure 1 and Figure 2) [11,15].

However, we noticed poor and/or ubiquitous, diffuse signals in six patients with histopathologically proven liver cirrhosis. One patient had received transarterial chemoembolization prior to hepatic resection. In this patient, no ICG staining appeared in the expected tumor area. Even in the three other patients without histopathologically proven liver cirrhosis, no staining signal was noticed. One of these patients developed a cardiac condition prior to the initial surgery and had received ICG nine days prior to the actual surgery; this may explain the lack of staining. In these patients, ICG was not considered helpful. In patients who suffer from later histopathologically confirmed cholangiocellular carcinoma (CCC), we noticed diffuse staining as well, which was not considered helpful.

However, in one patient, ICG signals led to the resection of two extra nodules. The histopathological analysis later confirmed the presence of a tumor in these extra nodules. Of note, in three cases, the persisting ICG staining after resection of the tumor led to a reresection in this area until no further staining was detectable, finally achieving histopathologically proven macro- and microscopically tumor residual (R0) resections in all three patients. 

Despite this, as there is still disagreement with regard to the ICG staining and resection margin, we performed a special analysis of this area by extracting samples of the rim of the staining signal alone. The histopathological analysis confirmed proof of the tumor-free resection margins in these patients (Figure 3).

## 4. Discussion

There are three main goals that should be aimed for when performing oncological liver resections, including achieving R0 resections, preserving healthy liver tissue and reducing postoperative complications [9,10,14].

In this single-center perspective study, when comparing ICG-supported RALS and RALS without intraoperative ICG staining, firstly, we could not identify any ICG-related adverse events. Secondly, no surgical or oncological inferiority was observed in the ICG cohort. Contrastingly, ICG-supported procedures lasted shorter and more R0 resections were achieved. ICG was considered helpful when performing surgery in the majority of cases, especially in combination with an IOUS. The combination enables the performing surgeon to gain real-time imaging with two different tools, thereby achieving safe oncological results, as well as parenchyma-sparing hepatic resections in different types of tumors without increased risks of complications. Additionally, ICG is not very cost-effective and the preoperative application only needs little organizational effort without requiring further equipment in case of an integrated (FireFly^TM^ camera Intuitive Surgical Inc.^®^, Sunnyvale, CA, USA) [14,24].

According to the updated Southampton guidelines, hepatic surgery should preferably be performed with a minimally invasive approach [1,25,26,27,28,29]. In this context, RALS was first reported by Giulianotti et al. in 2003 and is gaining more and more popularity [17,26,27,28,29].

RALS has been proven to be safe and non-inferior in comparison to open and laparoscopic surgery, including both minor and major hepatectomies [1,2,3,26,28,30,31,32]. However, long procedure times are often considered a drawback of robotic surgery [1,2,3,26,28,30,31,32]. They expose the patients to a different level of stress, i.e., from an anesthesiological point of view, causing further morbidity [6,33,34]. Overall, procedure times reported herein were comparable to the times reported by other studies [24,32,35]. Interestingly, procedure times for ICG-supported RALS were significantly shorter. This is potentially attributable to smaller tumors and, in turn, smaller resections, but also to a higher level of intraoperative security, due to the intraoperative ICG. ICG staining helps the surgeon to obtain real-time imaging of the extent of the tumor within seconds without switching instruments at the cost of valuable intraoperative time. Both tools help progress with faster intraoperative decision-making. On the other hand, ICG allows for a more straight forward approach, due to the marked resection margin [13,18,24]. Longer procedure times in the HC may also be attributable to the learning curve, which is always noted for a new technique and cannot be fully excluded in this study [36,37]. 

Segments resected during ICG-supported RALS include more segments, which are usually considered difficult to reach. This mirrors the superiority of the robot in narrow spaces, especially compared to laparoscopic procedures, taking advantage of the superior visualization and dexterity [7,24,27,28,38]. ICG potentially supports the performance of precise surgery in these narrow spaces. Very few intra- and post-operative complications were noted in both cohorts. This may be a reflection of the more precise and less harmful surgical approach, which improves the short-term outcome and quality of life [1,6,7,12,35,39,40].

From an oncological perspective, R0 resections and maximal parenchyma preservation are essential for the oncological outcome, especially when additional therapies or further resections may be required [13,35,41]. Tumors resected by ICG-supported RALS were significantly smaller, with a shorter distance from the resection margin. The average sizes of the tumors were 27.1 ± 25.0 mm and 47.6 ± 35.2 mm, respectively. Li et al. reported safe RALS in patients with tumors of a size of 5.6 cm, while Duong et al. only included tumors < 50 mm in diameter, both achieving good surgical and oncological results [6,42]. The distances to the resection margins were 5.8 ± 10.9 and 10.2 ± 13.4 mm, respectively, which is in line with the reports of others. [18,24,28,43]. Although the distances to the resection margins were longer in HC, a higher number of R0 resections were achieved by ICG-supported RALS. Higher R0 resection rates and increased oncological quality have been reported after ICG-supported RALS [7,12,13,41,44]. The consensus guidelines for the use of fluorescence imaging in hepatobiliary surgery, therefore, consider ICG helpful when choosing the right dissection plane, making the resection oncologically safer and parenchymal-sparing [13].

To perform parenchymal-sparing resections, as well as to achieve intraoperative security and R0 resections, the use of an IOUS is highly recommended when performing liver surgery [1,9,45]. While IOUS comes with known limitations, intraoperative ICG staining may help confirm real-time visualization of tumor and resection margins. One major drawback of ICG is still the lack of quantification of the signal, as well as the uncertainty of time and dose of application [14]. Varying doses and time points have been suggested and discussed by different authors, yet without gaining consent or a clear recommendation [11,14,17,18,19,46]. Patients included in this study were treated with 0.31 ± 0.06 mg/kg ICG 19:31 ± 24:21 h before the surgery. This discrepancy in time partly results from organizational, as well as logistic, in-house procedures. However, Franz et al., Alfano et al., Marino et al., and others recommend doses between 0.05 mg/kg and 0.5 mg/kg [11,12,17,19,20,24,41,44,47]. Takahashi et al., on the other hand, applied a dose of 2.5 mL/dL 2 to 1 day(s) before the surgery [44]. Interestingly, they experienced a wash out of ICG if injected more than 24 h before surgery, and therefore favor 24 h between injection and surgery [44]. This is contrary to the initial reports by Ishizawa et al. and others who warrant application times for up to 14 days before the surgery [6,11,19,20,46]. Kobayashi et al. further proposed an additional injection one day before the surgery if the initial application was too long before the surgery [48]. In contrast, Achterberg et al. report 10 mg of ICG 24 h before the surgery for patients who suffer from CRLM, while Liu et al. used a dose of 0.25 mg/kg 48 h before the surgery, achieving good staining results for FNH in atypical liver resections [18,49]. However, the recently published consensus guidelines recommend 0.5 mg/kg of body weight administered 10–14 days before the surgery [13]. However, if only administered 1–2 days before the surgery, 0.2 mg/kg should be applied, with special attention to patients who suffer from liver cirrhosis or post chemotherapy [50]. Wakabayashi et al., on the other hand, conclude in their recently published review regarding different doses that the identified doses were between 2.5 mg/body and 25 mg/body, applied within a range of three days before the surgery [14]. An additional problem is the relative novelty of ICG-guided liver surgery and the limited experiences, which make a clear recommendation difficult [14]. Owing to the heterogeneous study populations included in their studies, as well as the lack of systematic studies, Wakabayashi et al. suggest a patient-centered, tailored approach, by considering the age, underlying liver disease and the type of near-infrared camera before applying ICG [14].

Unlike others, we not only included patients who suffer from HCC and CRLM, but also patients who suffer from tumors and metastasis of different entities and found ICG-supported RALS in these patients to be safe and feasible from an oncological point of view, while being safe and especially parenchyma-sparing. However, most reports on ICG-based liver resections only include selected HCC and CRLM [6,10,11,19,24,42]. Rocca et al. reported the RALS of CRLM in a large series and considered it a safe procedure for these entities [35]. Marino et al. report the RALS for primary HCC, as well as CRLM, thus achieving both oncologically and surgically safe results [24] We can conclude that ICG-supported RALS not only works in these entities, but also in the metastasis of different origins. Unfortunately, we found no or diffuse staining, which implicates positive staining/results in patients who suffer from confirmed liver fibrosis or advance stages of cirrhosis. Further studies have already described this aspect [8,10,12,17]. This may be due to impaired up-take and excretion mechanisms in fibrotic and/or cirrhotic liver tissue [12]. 

Of note, ICG is not only limited to the aforementioned indications. Originally, ICG was used to test liver function, liver and biliary anatomy, tumor location or biliary leakages [13,51,52]. It is further used to define resection margins for oncological resections, but also resection planes for living donor liver transplantations [13]. Additional indications for the use of ICG are bleeding or perfusion control, i.e., in esophageal or colorectal surgery, and lymph node detection [13,52,53,54,55].

However, this study has to deal with the limitations, firstly, of a partly retrospective study and, secondly, the limited number of patients included in this small clinical study. Additionally, collectives consist of heterogeneous study populations. Furthermore, due to the study design, some procedures may fall into the time of the learning curve of performing robotic surgeons. As mentioned, there is still a certain insecurity regarding the use and application of ICG, as well as the quantification of the signal. Furthermore, we cannot provide valuable results for long-term outcomes, as the patients who are recently included only have a short follow-up period.

## 5. Conclusions

We conclude that ICG is a safe, helpful intraoperative tool when performing ICG-supported RALS in selected cases, i.e., HCC or LM of different origins, for different reasons. ICG provides additional, intraoperative real-time imaging in addition to IOUS. This enhances the intraoperative surgical precision and decision-making by meeting the initially stated requirements of RALS, including oncological (R0 resection) and surgical safety (reducing postoperative complications), while being precise (preserving healthy liver tissue,) without prolonging surgery or compromising the surgeon’s performance [13,14,42]. ICG-supported RALS is a safe procedure, especially in smaller tumors of different origins. We, therefore, warrant preoperative ICG application in cases of scheduled RALS.

## Figures and Tables

**Figure 1 jcm-11-03527-f001:**
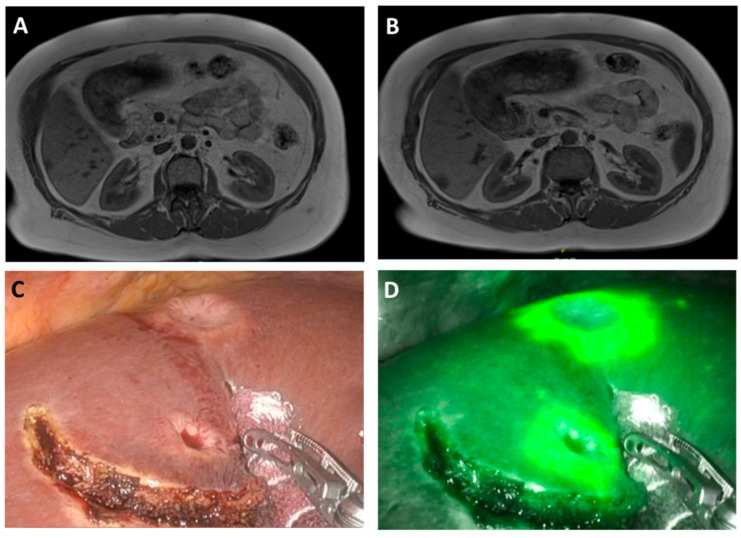
Intra-operative robotic image in white light (**C**) and intraoperative near-infrared (**D**) image of CRLM (rim phenomenon) with corresponding preoperative T-weight MRI-scans (**A**,**B**)

**Figure 2 jcm-11-03527-f002:**
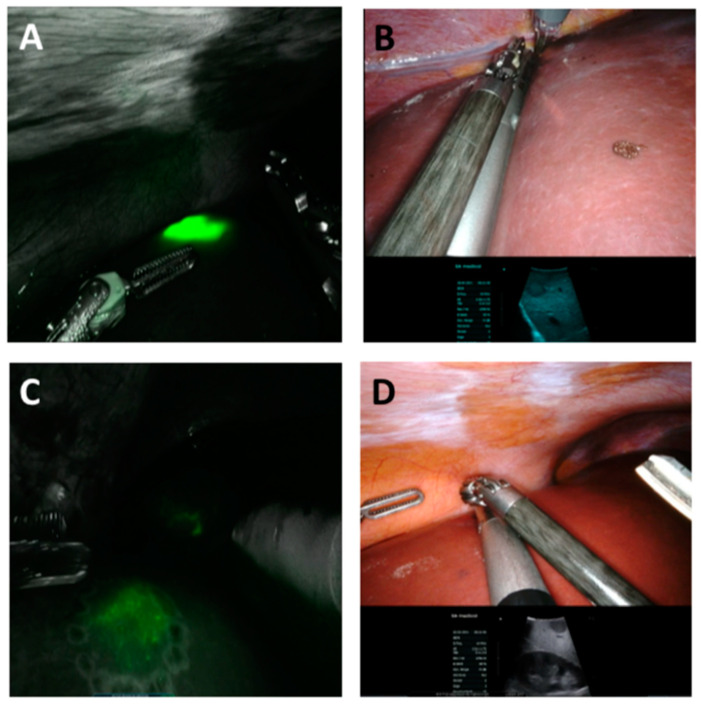
Intra-operative near-infrared image (**A**,**C**) and normal white light image (**B**,**D**) with the corresponding image of the IOUS of unequivocal staining (**A**,**C**) of a HCC, corresponding with the IOUS (HCC) and diffuse staining and not corresponding with the ICG-based image (**C**,**D**) of a CCC.

**Figure 3 jcm-11-03527-f003:**
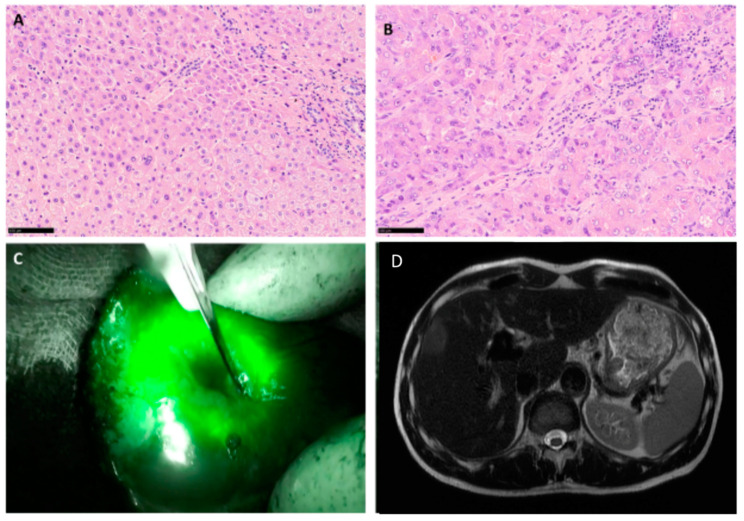
Intra-operative fluorescence imaging and histopathological image of the corresponding area confirming R0 resection. Histological images of tumor free liver tissue in the marginal area with portal fields and a sparse round cell infiltrate (**A**) HE staining, magnification 200×. Histological images of HCC with trabecular and pseudoglandular growth from the same patient (**B**) HE staining, magnification 200×. Intraoperative imaging of the fluorescent tumor after ICG application (**C**), preoperative MRI scan, showing the lesion (**D**).

**Table 3 jcm-11-03527-t003:** Operative data of the comparison of patients who underwent RALS with and without obtaining preoperative intravenous ICG.

	Total (*n* = 54)	ICG (*n* = 28)	HC (*n* = 26)	*p*-Value *
**Operative Data**				
Type of resection				
Wedge resection	34.0	40.7	26.9	0.513 ^b^
Segment resection	35.8	44.4	26.9	0.221 ^b^
Lobectomy	5.7	0.0	11.5	0.135 ^b^
Left hemihepatectomy	18.9	14.8	23.1	0.626 ^b^
Right hemihepatectomy	5.7	0.0	11.5	**0.031** ^b^
Segment resected				
I	1.0	2.1	0.0	0.331 ^b^
II	19.4	17.0	21.6	0.178 ^b^
III	20.4	14.9	25.5	0.181 ^b^
IV	10.2	6.5	13.7	0.249 ^b^
V	16.3	17.0	15.7	0.675 ^b^
VI	14.4	17.0	11.8	0.637 ^b^
VII	5.1	8.5	2.0	0.186 ^b^
VIII	13.3	17.0	9.8	0.150 ^b^
Length of surgery (min) mean ± SD	192.3 ± 97.7	142.7 ± 61.8	246.4 ± 98.6	**<0.001 ^a^**
Intra-operatively realized complications (yes) %	5.6	10.7	0.0	0.086 ^b^
Simultaneous CHE (yes) %	40.4	25.0	53.8	**0.030** ^b^
Intra-operative placement of a drain (yes) %	75.9	71.4	80.8	0.422 ^b^

Data are presented as mean ± standard deviation (SD), range, or relative frequencies. Continuous variables were tested using ^a^ Students’ *t*-test (normally distributed), while categorical variables were compared using ^b^ chi-square; values in bold were considered statistically significant (*p* < 0.05). “* *p*-values” refer to values in the middle and right column. ICG: indocyanine green; CHE: cholecystectomy; HC: historic cohort and SD: standard deviation.

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
