# Peer review of "The Role of ICG in Robot-Assisted Liver Resections"

_jcm, 2022, doi:10.3390/jcm11123527_

Round 1
Reviewer 1 Report
Thank you for the opportunity to review the manuscript about robotic liver surgery supported by ICG application. This paper displays major advantages and disadvantages of ICG-RALS and is of importance fort the further use of this novel surgical tool. The Manuscript has been written with great enthusiasm. Nonetheless, I have a few ideas that might improve the manuscript.
General Comment:
In a lot of phrases the sentences are built pretty complicated, a lot of insertions, that hinder the readability of the phrases, although the general quality of language is pretty good. A careful read through would be advisable, as several little spelling mistakes occur every now and then.
Introduction
I suggest that the abbreviations used in the introduction should be written in full at first mention (such as IOUS).
Methods
Was a specific questionnaire used after surgery to evaluate usefulness of ICG application/staining? At least from your results section it seems like.
Results
The most resected pathologies should be put in order according to frequency to provide better readability of the table. Would it be possible to split the table in different tables to increase the possible overview.
Discussion
I would recommend straightening the discussion section to point out the messages of the specific paragraphs more concisely.
Author Response
Dear Dr. Vernadakis,
Dear Reviewer
Thank you for considering our revised manuscript titled The role of ICG in robot-assisted liver surgery’ (jcm-1770554) for re-submission in your Special Issue on “Overcoming the Barriers of Minimally Invasive Liver Resection: Future Persepectives, Challenges and Limitations”.
Firstly, we should like to thank you and the reviewer for perusing our manuscript and for the additional excellent and helpful remarks. We appreciate the reviewers’ comments and welcome the opportunity to address these queries and concerns in our revised manuscript. We have addressed below all the reviewers’ comments in our responses to both yourself and the reviewer. We now re-submit to you a revised version of our study, along with this cover letter, which answers the reviewers’ questions and criticisms, in seriatim. The changes made in the revised manuscript are highlighted in yellow for clarity.
We hope that the revised manuscript, together with this point-by-point discussion, will now address all concerns and render it suitable for publication in your special issue on “Overcoming the Barriers of Minimally Invasive Liver Resection: Future Persepectives, Challenges and Limitations Langenbeck’s Archives of Surgery”.
Sincerely,
Anne-Sophie Mehdorn

Reviewer 2 Report
The points of this paper are to assess whether the use of intra-operative ICG fluorescence imaging could improve the feasibility of robot-assisted liver resections (RALS) and could enhance surgical and oncological outcomes. Another endpoint of this study is to evaluate whether the use of ICG is associated with an increased occurrence of side effects.
This paper presents a retrospective monocentric study comparing a group of patients who underwent an ICG assisted RALS to a historical cohort of patients who underwent a RALS. This study showed that ICG assisted RALS are feasible, with a shorter operating time, a tendency to less R1 resection. It also showed a non-inferiority of this technique for surgical and oncological outcomes. No increase in side effects occurrence was found.
This is an interesting study evaluating the efficiency of a now commonly used imaging modality in improving minimally invasive liver surgery. Some points need to be clarified.
Comments
1/ A significant issue of this paper is its historical bias. The outcomes of subjects who underwent ICG assisted RALS between 2019-2021 were compared to those who underwent standard RALS between 2014-2019. Knowing how long the learning curve in the robotic assisted surgery practice is, how can we be sure that the results observed for the shorter operating time and other surgical outcomes are not the effect of an improvement in the surgeons’ technique?
2/ The authors would improve the quality of this article by adding more precision to the presented data. For instance, the ICG fluorescence technique is associated to intra-operative ultrasound system in 90% of the cases, but the authors show a correlation of the results found with both those techniques in 81%. What is the reason of the correlation in only 81%?
3/ The authors may develop a paragraph about the other ways to use the ICG in minimally invasive liver surgery. It is well understood that the present use of ICG is the intra-operative detection of liver tumors, but other applications could be evaluated. Those applications are well described in the recent international guidelines for the use of ICG (Wang, 2021). Even if the study is not designed to evaluate them, the authors could mention them to scan all the ways this new imaging technique could improve liver resections.
Author Response

(The authors gave the same response as above.)

Reviewer 3 Report
The work entitled: The role of ICG in robot-assisted liver resections, describes the comparative advantages in liver surgery, using robotic assistance, of the use of Indocyanine green dye versus the historical record without its help.
The work is well written, but it is based on a variety of cases that are not always comparable due to the small population; for example, patients with the previous cholecystectomy it does not have similar populations nor with cystoprostatectomy.
However, the comparisons with other procedures are adequate and correctly mark the point that tries to prove the advantage of using the stain.
Other published works point in the same direction as this work, so much of the insights that it pours into the discussion are concurrent with what has been published.
It requires correcting minor points that are listed:
1) The initials HCC, FNH are defined up to the foot of the figure in table 1; you must define their acronyms the first time they are used
2) The acronym IOUS (intraoperative ultrasound) does not define it
3) The acronym R0 (no macroscopic or microscopic residual tumor) does not define it
Author Response

(The authors gave the same response as above.)
